# The Mitochondrial Small Heat Shock Protein HSP22 from Pea is a Thermosoluble Chaperone Prone to Co-Precipitate with Unfolding Client Proteins

**DOI:** 10.3390/ijms21010097

**Published:** 2019-12-21

**Authors:** Marie-Hélène Avelange-Macherel, Aurélia Rolland, Marie-Pierre Hinault, Dimitri Tolleter, David Macherel

**Affiliations:** IRHS, Agrocampus-Ouest, INRA, Université d’Angers, SFR 4207 Quasav, 42 rue George Morel, 49071 Beaucouzé, France; marie-helene.macherel@agrocampus-ouest.fr (M.-H.A.-M.);

**Keywords:** desiccation tolerance, heat stress, mitochondria, seed development, small heat shock proteins

## Abstract

The small heat shock proteins (sHSPs) are molecular chaperones that share an alpha-crystallin domain but display a high diversity of sequence, expression, and localization. They are especially prominent in plants, populating most cellular compartments. In pea, mitochondrial HSP22 is induced by heat or oxidative stress in leaves but also strongly accumulates during seed development. The molecular function of HSP22 was addressed by studying the effect of temperature on its structural properties and chaperone effects using a recombinant or native protein. Overexpression of HSP22 significantly increased bacterial thermotolerance. The secondary structure of the recombinant protein was not affected by temperature in contrast with its quaternary structure. The purified protein formed large polydisperse oligomers that dissociated upon heating (42 °C) into smaller species (mainly monomers). The recombinant protein appeared thermosoluble but precipitated with thermosensitive proteins upon heat stress in assays either with single protein clients or within complex extracts. As shown by in vitro protection assays, HSP22 at high molar ratio could partly prevent the heat aggregation of rhodanese but not of malate dehydrogenase. HSP22 appears as a holdase that could possibly prevent the aggregation of some proteins while co-precipitating with others to facilitate their subsequent refolding by disaggregases or clearance by proteases.

## 1. Introduction

In all organisms, the homeostasis of proteins relies on several families of molecular chaperones that intervene in biogenesis, translocation, folding, and assembly of polypeptides to survey their structure and function in the crowded environment of cells [1]. Under stress conditions leading to protein structure destabilization, the role of molecular chaperones becomes critical, and their expression often increases massively to prevent unfolding and contribute to the disaggregation and refolding of client proteins. Because molecular chaperones figure among proteins heavily induced by heat shock, they have been coined HSP (heat shock protein), although the name does not correctly reflect their primary function [1].

The small HSPs (sHSPs) are among the most widespread but less conserved family of molecular chaperones (for a recent review, [2]). Over 8000 sHSP sequences have been identified in organisms belonging to all kingdoms [3], and a database (sHSPdb) has been established [4]. sHSP proteins range in size from 12 to 42 kDa and generally assemble into multimers of 12 to over 32 monomers [5]. All sHSPs share a 90-amino acids beta-sheet domain called alpha-crystallin domain (ACD), by reference to alphaA- and alphaB-crystallin of mammalian eye lens [6]. The ACD is followed by a non-conserved short C-terminal extension, which is involved in sHSP oligomerization [7]. A variable N-terminal region forms a disordered and flexible arm, which interacts with protein substrates [8]. There is mounting evidence that ACD, N-, and C-terminal regions contribute all together to sHSP structure and molecular chaperone activity [3,9]. sHSPs have been suggested to bind denatured proteins in an ATP-independent manner, limiting aggregation and allowing subsequent refolding in cooperation with other chaperones, such as HSP70 [1,10]. sHSPs were qualified as holdase chaperones by contrast to foldases that assist protein folding by ATP-dependent mechanisms [11]. Most sHSPs are strongly induced upon heat shock and more generally by abiotic stress [2,12,13].

Although they are almost ubiquitous, sHSPs are especially prominent in plants [3] with, for instance, 19 sHSP genes identified in *Arabidopsis thaliana* [14] and 94 in cotton [15]. Based on their intracellular localization, sequence homology, and immunological cross-reactivity, 11 subfamilies of plant sHSPs were defined: six nuclear/cytoplasmic localized subfamilies (CI to CVI) and five organelle localized subfamilies (for review, see [16]. Several reports showed that sHSP overexpression could protect microorganisms, animals, and plants from heat and/or oxidative stresses [17,18,19,20]. Conversely, sHSP deficiency was shown to decrease stress tolerance [21,22]. The role of sHSPs has also been investigated in the context of cold and freezing stress or water deficiency [17,19,23].

While sHSPs are generally highly stress-inducible, several were also shown to accumulate in the absence of environmental cues in animals and plants. For instance, alpha crystallins represent more than 50 % of vertebrate lens proteins [6]. In *Drosophila melanogaster*, the mitochondrial HSP22 is constitutive but up-regulated during aging [24,25]. In plants, a constitutive expression of sHSPs in vegetative tissues has been reported in the resurrection plant *Craterostigma plantagineum* [26] and in chestnut stem [23]. However, plant sHSP accumulation occurs mainly under specific developmental stages such as pollen and seed formation or fruit maturation [26,27,28]. The heat-independent expression of seed sHSPs is triggered by HSFA9, a specific member of the heat transcription factor HSF family [29,30]. HSFA9 appeared itself under the control of abscisic acid (ABA) via the master seed transcription regulator ABI3 [30]. Manipulating HSFA9 expression highlighted the overall importance of sHSPs for desiccation tolerance and seed longevity [31,32]. More recently, [33] demonstrated that overexpressing a rice sHSP in *Arabidopsis* seeds improved seed vigour and longevity and was beneficial for seedling establishment under stress conditions.

The effect of heat shock on sHSPs oligomerization has been widely investigated in microorganisms [34], animals [35], and plants [5,36]. Different mechanisms of heat-induced activation have been proposed: dissociation of sHSP multimers into smaller species (monomer to tetramers), increased rates of subunit exchange, as well as other conformational changes [5]. Activated sHSPs may bind to hydrophobic patches exposed at the surface of denatured proteins, thus preventing aggregation and facilitating further disaggregation of the sHSP co-aggregates by dedicated molecular chaperones [37]. Preferred clients have been suggested for sHSPs [38,39], and variable chaperone activity among related sHSP was observed by *in vitro* assays. For instance, [40] demonstrated in pea, *Arabidopsis*, and wheat that cytosolic class II sHSPs were more effective than cytosolic class I sHSPs to protect luciferase from heat aggregation. The protection of membrane protein complexes (Mitochondrial complex I, Photosystem II) by sHSPs was also demonstrated during heat, salt, and heavy metal stress [41,42,43,44]. In addition, sHSPs may also contribute to membrane protection upon stress by binding membranes and preserving their fluidity [45,46,47,48,49]. The cyanobacterial HSP17 was shown to behave as an amphitropic protein by protecting both proteins and membranes during heat shock [50].

Genomic data support that mitochondrial sHSPs are ubiquitous in plants but seldom found in other eukaryotes [5]. An exception is the mitochondrial sHSP of the fruit fly *Drosophila melanogaster*, which accumulates during stress and aging [18]. This sHSP was shown to protect mitochondria against oxidative damage and be a critical component of fly longevity [18,21,24,25]. It was also proposed to be involved in the protein unfolding response [24]. In plants, only few mitochondrial sHSPs have been investigated thus far [51,52,53,54,55]. The pea HSP22 was originally discovered in pea vegetative tissues exposed to heat stress [51,52]. This protein was found exclusively within mitochondria, where it represents 1–2% of total matrix proteins. A plant mitochondrial proteomic survey further revealed a specific accumulation of HSP22 in pea seeds in the absence of environmental stress, showing that HSP22 expression was also developmentally regulated [56]. The majority of angiosperm seeds are desiccation tolerant, which allows seeds to be stored for long periods in the dry state, a property that contributed to the evolutionary success of angiosperms and the development of agriculture. Together with late embryogenesis abundant proteins (LEA), sHSPs are generally expressed during late seed development and are thus expected to be involved in desiccation tolerance [57,58,59,60]. The protection of mitochondria in dry seeds appears to be of primordial importance because of their bioenergetic role, which has to resume very early during imbibition to fuel metabolism [61,62]. Besides HSP22, pea seed mitochondria were also found to accumulate in their matrix LEAM, a LEA protein shown to protect the inner membrane during desiccation [63,64]. The remarkable stress tolerance of pea seed mitochondria that still perform oxidative phosphorylation at -3.5 or 40 °C was correlated with the presence of HSP22 and LEAM [65]. Both proteins were detected with similar abundance in the seeds of 91 different accessions of pea, which underlies their importance for the protection of mitochondria [66].

Here, we investigated the structural features and the molecular functions of HSP22 in relation with stress tolerance using recombinant proteins and a wide array of biophysical and biochemical tools. HSP22 proved to share common features with plant and non-plant sHSPs already described: ß-strand enriched secondary structure, large multimers that dissociate into sub-oligomers upon heating. The results highlight the remarkable thermostability of the protein together with its ability to co-precipitate with unfolding client proteins. This strongly favors a role of HSP22 in the association with aggregated proteins, presumably to facilitate their refolding during stress recovery.

## 2. Results

### 2.1. Structural Features of Recombinant HSP22 (HSP22rec) as a Function of Temperature

We first examined whether the structure of the recombinant HSP22 (HSP22rec) could be affected by an increase of temperature from 20 °C to 42 °C, which corresponds to severe heat stress conditions for most plants. Overall, HSP22rec secondary structure at 20 °C comprises 6% alpha helix, 36% beta strand, 20% turn, and 38% random coil (Figure 1). These results are consistent with the typical tertiary structure of sHSPs that harbour a conserved ß sandwich ACD and a flexible N-terminal arm. We could not detect any major change in the secondary structure of the protein at 42 °C, which indicates that the overall structure of the polypeptide is not affected by this large increase in temperature.

Although HSP22 tertiary structure has not been experimentally resolved yet, its amino acid-sequence is highly similar to those of two plant sHSPs whose structures have been determined: HSP22 shares 35% identity (63.5% similarity) with the *Arabidopsis* chloroplastic AtHSP21 [67] and 26.5% identity (49% similarity) with the wheat TaHSP16.9 [36]. Unsurprisingly, the higher conservation was found in the ACD domain (Appendix A). In order to confirm that HSP22 should adopt the typical tertiary structure of sHSPs, we used the modeling software Swiss Model [68] to predict its 3D structure. Using AtHSP21 as a template, the model for HSP22 displays the typical architecture of sHSP with a beta–sandwich formed by anti-parallel beta strands in the ACD domain and less ordered N and C termini (Appendix A).

Since dynamic oligomerization of sHSPs is of paramount importance for their chaperone function, we analyzed the quaternary structure of HSP22rec using size-exclusion chromatography at 20 °C, and at 42 °C to mimic severe heat stress conditions. At 20 °C, HSP22rec was found mainly in high molecular mass fractions corresponding to 20 mers, while a smaller proportion of 12 and 8 mers could be detected (Figure 2). These results suggest the presence of polydisperse oligomers of HSP22rec at ambient temperature. When size-exclusion chromatography was performed at 42 °C, high molecular structures were no longer observed (Figure 2). At high temperature, HSP22rec was found in the fractions corresponding to the monomeric form.

Collectively, these results demonstrate that heat stress conditions do not interfere with HSP22 polypeptide structure but cause the large oligomers to preferentially dissociate into monomers. This dissociation was reversible, as heated and non-heated HSP22rec displayed the same molecular mass in native PAGE (Polyacrylamide gel electrophoresis), which was run at 20 °C (Appendix A). Such a reversibility of thermal dissociation was reported earlier for other sHSPs [36,69,70]. Since exposure of hydrophobic patches was proposed to trigger interaction between sHSP and unfolding proteins [40], the surface hydrophobicity of HSP22rec and two other mitochondrial proteins, rhodanese and fumarase, was examined using saturating concentrations of bis-ANS (bis-8-anilinonaphthalene-l-sulphonic acid), a compound that displays increased fluorescence while bound to protein hydrophobic sites. The bis-ANS probe was added immediately after incubating the proteins at 23 °C or 55 °C for 15 min. As shown in Figure 3, the fluorescence of bis-ANS increased strongly when rhodanese and fumarase were incubated at 55 °C but not in the case of HSP22rec, for which bis-ANS fluorescence remained unchanged. This supports our observations on the intrinsic structural stability of the HSP22 polypeptide and suggests that exposure of hydrophobic patches would rather concern unfolding client proteins than sHSP themselves.

The thermosolubility of HSP22 was further investigated by heating the recombinant protein at high temperatures (50 to 90 °C). The presence of HSP22rec in soluble and insoluble fractions was determined by SDS-PAGE after centrifugation of the samples. As shown in Figure 4, the protein remained fully heat soluble up to 90 °C. The small amount of HSP22rec found in the pellet at 50, 70, and 90 °C was likely due to contamination by supernatant. We also observed a significant amount of HSP22rec in the insoluble fraction when the protein was not heated (Figure 4). This was likely due to the formation of some aggregates during protein storage at -80 °C, which would be resolubilized at high temperatures. As a whole, these results highlight the heat structural stability (and solubility) of the HSP22rec polypeptide, which dissociates from large oligomers to monomers upon heating.

### 2.2. Chaperone Assays with Single Mitochondrial Proteins

Several sHSPs were shown to contribute to the cellular chaperone network by passively binding to hydrophobic exposed regions of partially unfolded proteins. This binding decreases further aggregation and maintains a folding-competent state even when proteins are sequestered within aggregates [71]. To investigate the molecular function of HSP22 *in vitro*, chaperone assays combining HSP22rec and commercially available mitochondrial proteins (rhodanese, malate dehydrogenase) were carried out. As a control, bovine serum albumin (BSA) or lysozyme were used instead of HSP22 in aggregation assays at 50 °C. At this temperature, HSP22, BSA, and lysozyme remained thermosoluble in contrast to rhodanese and malate dehydrogenase (see Figure 4, Figure 5, Figure 6 and Figure 7, Appendix A). At 50 °C, neither HSP22rec nor BSA could prevent aggregation of rhodanese at a mass ratio of 1:1 (Figure 5). This corresponded to a 1.6:1 molar ratio of HSP22rec to rhodanese and to a 0.5:1 molar ratio of BSA to rhodanese. Although we showed above that HSP22rec was heat soluble, approximately two thirds of the protein were found in the insoluble fraction with rhodanese. In contrast, BSA remained mostly soluble in the presence of aggregating rhodanese. The aggregation assay was repeated with malate dehydrogenase instead of rhodanese and led to similar conclusions (Appendix A). This strongly suggests that HSP22rec binds to heat denatured rhodanese or malate dehydrogenase polypeptide and co-precipitates in aggregates.

At a higher molar ratio (6:1), HSP22rec was able to partially protect rhodanese from heat aggregation (Figure 6a). Interestingly, HSP22rec amount in the insoluble fraction correlated with the extent of rhodanese aggregation. The effect of lysozyme was also studied to take into account a possible implication of macromolecular crowding on rhodanese heat aggregation in our assay conditions. Lysozyme, which is a small thermostable protein like HSP22, was not effective in protecting rhodanese from heat-induced aggregation (Figure 6b), suggesting a specific chaperone effect of HSP22rec. Measurements of rhodanese activity, which was very sensitive to the heat treatment (90% loss of activity), showed that a nine-fold ratio of HSP22rec, which can partly prevent aggregation, was unable to preserve the enzyme activity (Appendix A).

The thermal aggregation of malate dehydrogenase at 50 °C was also studied in the presence of three, six, or nine molar excess of either HSP22rec or lysozyme (Figure 7). A small amount of malate dehydrogenase remained soluble when the enzyme was heated alone or with HSP22rec, indicating no significant protective effect of HSP22rec (Figure 7a). Surprisingly, in the presence of nine molar excess of lysozyme, malate dehydrogenase fully aggregated upon heating (Figure 7b). Compared to the rhodanese aggregation assay (Figure 6a), more HSP22rec was found in the insoluble fraction when heated in the presence of malate dehydrogenase (Figure 7a). This result further supported the co-precipitation of HSP22rec with aggregating client proteins.

These chaperone assays with individual proteins indicate that HSP22rec, even at high molar ratios, has only modest but selective protective effects against the aggregation of putative client proteins, but in all cases co-precipitates with aggregated proteins. To study the function of HSP22 in complex protein environments, we then examined its properties and effects in overexpressing bacteria and mitochondrial matrix.

### 2.3. Effects of HSP22 in Bacteria and Complex Protein Extracts

The protective effect of HSP22 was evaluated *in vivo* by exposing HSP22 overexpressing bacteria to a severe heat shock (48 °C for 30 min) and then monitoring their survival. Non-induced bacteria were used as a control in this experiment, and bacteria were grown at 30 °C in order to increase the threshold between growth and heat shock temperature. The recombinant protein was strongly expressed since, after 1 h of induction, it was the major band in a total protein extract analyzed by SDS-PAGE (Appendix A). The survival after heat shock was significantly higher (on average twice) for induced bacteria compared to non-induced bacteria (Figure 8), which demonstrates that the accumulation of HSP22 in bacteria increases their thermotolerance.

When protein extracts from bacteria overexpressing HSP22 were heated at increasing temperatures, HSP22rec started to precipitate at 60 °C as well as most of the *Escherichia coli* proteins (Figure 9). This indicates that HSP22rec co-precipitates with aggregated proteins in a complex bacterial extract as well as in aggregation assays with single proteins.

Since HSP22 naturally accumulates to a high level in mitochondria of pea seeds [56], we performed a similar assay of thermosolubility using a soluble extract of mitochondrial proteins. When the extracts were heated from 45 to 90 °C, most proteins started to precipitate at 55–60 °C, the native HSP22 following a similar pattern (Figure 10a), although a significant amount of the protein remained soluble above 60 °C, as confirmed by immunodetection (Figure 10b).

Size-exclusion chromatography was also performed at 20 °C and 42 °C using the soluble protein extract from pea seed mitochondria. At 20 °C, the native HSP22 was retrieved only in high and low molecular mass fractions (Figure 11). The native protein was preponderant in the fractions corresponding to 163 to 422 kDa (8 to 21 mer), while a smaller proportion of the protein was found as dimers and trimers. These values were consistent with the polydisperse large oligomers observed for HSP22rec at 20 °C. When size exclusion chromatography of the mitochondrial soluble protein extract was performed at 42 °C, the elution profile was not markedly affected by temperature, although a greater abundance of larger oligomers and of dimers and monomers was observed at 42 °C compared to 20 °C (Figure 11). This contrasts with the situation of the purified HSP22rec, which was found almost exclusively monomeric at 42 °C (Figure 2), showing that temperature effect on HSP22 oligomerization is strongly dependent on the protein environment.

Finally, since HSP22 accumulates in seed mitochondria together with LEAM, which was shown to protect the inner membrane upon drying [63,64], we examined whether the protein could also contribute to membrane stabilization upon drying. Indeed, several sHSPs were shown to interact with membrane phospholipids to maintain fluidity upon stress conditions [50]. A liposome integrity assay allowing us to monitor the leakage of a fluorochrome after drying and rehydration of liposomes was used, adding recombinant LEAM, fumarase, or HSP22 before drying. Neither HSP22 nor fumarase—which was used as a negative control—had any protective effect, while LEAM could partially maintain integrity of liposomes (Appendix A). This suggests that, in the context of seed desiccation tolerance, the role of HSP22 is not related to membrane protection.

## 3. Discussion

The interest for sHSPs is well illustrated by a query on PubMed (www.ncbi.nlm.nih.gov/pubmed/) using the string [small heat shock protein* OR sHSP*] that yields more than 2700 publications, among which ≈ 10% are reviews (as of October 2019). Although there is a consensus for a general role of these ATP-independent chaperones in the prevention of irreversible aggregation of unfolded protein, in cooperation with other chaperones, sHSPs are still considered as enigmatic proteins [72]. Here, we analyzed structural and functional properties of the pea mitochondrial HSP22, which is highly heat-inducible but also accumulates to a high level in seeds. 

Both the secondary structure analysis and the 3D modeling of HSP22 were consistent with data available for other sHSPs [5,36,73], which indicates that HSP22 harbors the canonical 3D structure of sHSPs. Most sHSPs form large polydisperse or homogenous oligomers with dimers as building blocks [16]. However, some sHSPs were found to be monomeric [74], dimeric [75], or tetrameric [76]. Using size exclusion chromatography at room temperature, both purified HSP22rec and native HSP22 within the mitochondrial soluble extract were found essentially as large polydisperse oligomers of 12 to 20 subunits. This is in agreement with the usual range of 12 to 32 subunits described for sHSPs [3] and with the 20 mer size recently reported for a sugarcane mitochondrial sHSP [55]. Smaller size structures such as dimers were also shown to coexist with HSP22 large oligomers at room temperature. The occurrence of dimers along with dodecamers was previously reported for the wheat HSP17.8 and the *Arabidopsis* HSP17.7 recombinant proteins when analyzed by nano-electrospray ionisation mass spectrometry [40].

Several reports on plant and animal sHSPs revealed that heat induced structural modifications and an increase in hydrophobic site exposure at elevated temperatures [35,40,74,77]. However, neither the secondary structure nor the hydrophobic site exposure of HSP22rec were affected by an increase from 20 to 42 °C. Interestingly, the secondary structure of the sugarcane mitochondrial sHSP was also maintained up to 42 °C [55]. The structural heat stability of HSP22 suggests that interactions with clients could be mediated by electrostatic forces and hydrogen bonding rather than by hydrophobic bounds [2]. Alternatively, HSP22 could expose more hydrophobic residues upon heating only in the presence of the unfolded substrates, as proposed for other sHSPs [78,79].

In contrast with this intrinsic structural thermostability, the oligomeric landscape of purified HSP22rec was highly sensitive to the increase in temperature, which caused almost full dissociation of the large oligomers into monomers. However, when the oligomerization state of HSP22 was analyzed within seed mitochondria matrix extract, heat did not cause such a strong shift from oligomers into monomers. Instead, at 42 °C, larger oligomers than those observed at 20 ° C were the most abundant form of HSP22. This result suggests that, in a complex protein extract, heat-destabilized proteins may associate with HSP22 suboligomers to form high molecular mass complexes, preventing further irreversible aggregation. Another explanation could be that, upon heat treatment, released monomers or dimers assemble with residual oligomers to form higher-order oligomers, as demonstrated for the pea cytosolic Hsp18.1 [70]. The self-association of transient monomers into sheet-like-supermolecular assemblies (SMAS) was also reported for *Caenorhabditis elegans* HSP17 [80]. However, such higher-order oligomers are rather unlikely for HSP22, since only monomers were detected when purified HSP22rec was heated at 42°C. Heat dissociation of sHSP oligomers was previously reported for other plant and non-plant sHSPs [2], generally yielding sHSP dimers [5]. Crystal structures further supported that dimers were the building block of most sHSPs [36]. Dimers were proposed to be the form that captures heat-sensitive unfolding proteins [5,81]. However, heat-induced dissociation of oligomers is not a general feature of sHSPs: for example, plant Class II cytosolic sHSP [40], yeast cytosolic HSP42 [34], sugarcane chloroplast, and mitochondria sHSP [55] remained oligomeric upon heat treatment. Size exclusion chromatography suggests that HSP22 large multimers are composed of an even number of subunits, which is consistent with the canonical dimer association. Dimers released upon heating of the purified HSP22 could not be observed at a significant level in our experiments, likely because of their dissociation into monomers. To our knowledge, there are few reports about the thermosolubility of purified sHSPs. In plants, Basha et al. [40] mentioned that the pea class II cytosolic PsHsp17.7 was heat-soluble at 42 °C. In animals, the human betaB2-crystallin aggregated above 50 °C while the calf ortholog remained soluble up to 90 °C [35]. Thus, thermosolubility at high non-physiological temperatures may not be a general feature of sHSPs. In the case of HSP22, the purified protein remained soluble at high temperature, but its solubility was strongly affected by surrounding proteins depending on their heat sensitivity. Indeed, HSP22 was found to precipitate in proportion to heat-denatured substrates. This property is consistent with the high affinity of sHSPs for unfolded proteins and their proposed involvement in protein aggregate re-solubilization [16,37,82]. However, this result contrasts with most chaperone assays, where sHSPs remained soluble when heated with other proteins whether these client proteins remained soluble or not [40].

The presence of sHSPs has been demonstrated to improve the thermosolubility of bacterial or yeast proteome [17,34]. In addition, this property was often correlated with a better stress tolerance when sHSPs were overexpressed in prokaryotes and eukaryotes [17,18,83]. However, HSP22rec did not change the protein aggregation onset temperature of *E. coli* soluble proteome, and we could not demonstrate any prevention of aggregation upon heat treatment. In line with this result, a sugarcane mitochondrial sHSP was not able to protect *E. coli* proteome from heat, in contrast to the plastidial sHSP [55]. In such *in vitro* experiments, the ineffectiveness of sHSPs could be due to the lack or the dilution of other factors (other chaperones, metabolites) required for heat protection. Constitutive and/or induced chaperones may act together with HSP22 to prevent irreversible protein aggregation or re-solubilize aggregated proteins. Indeed, the accumulation of HSP22rec appeared to be beneficial to *E. coli*, since more HSP22 overexpressing bacteria survived a sub-lethal heat shock than wild type bacteria. Previously, it was shown that oxidative phosphorylation of pea seed mitochondria was more efficient at high temperature than that of pea epicotyl mitochondria, which do not contain HSP22. Therefore, the expression of HSP22 appears to be beneficial with respect to heat tolerance, both for plant mitochondria and bacteria.

*In vitro* assays using light scattering or centrifugation to monitor aggregation complex formation provided evidence that sHSPs could prevent heat aggregation of specific client proteins. A concentration-dependent inhibition of protein aggregation has been often observed [73,74,84], but high molar excess of sHSPs were usually required to achieve protein protection. For example, citrate synthase and malate dehydrogenase were prevented from heat aggregation in the presence of 12 molar excess of the *Arabidopsis* chloroplast HSP21 [79]. This ratio was also required for AtHSP18.5 in order to maintain luciferase solubility, while only two-molar excess prevented malate dehydrogenase aggregation [75]. We carried out aggregation assays with several mitochondrial proteins that could potentially be clients for HSP22 in a natural context, but most proteins could not be protected by HSP22rec from heat aggregation, even when using nine molar excess of chaperone. In contrast with previous studies [8,40,84], HSP22 was found in the insoluble fraction with heat-denatured substrates, suggesting that the primary fate of the protein is to co-precipitate with heat-denatured proteins in order to facilitate further disaggregation by molecular chaperone unfoldase machineries [1,37]. The presence of a high molar ratio of HSP22 could effectively reduce aggregation of rhodanese but not of malate dehydrogenase. The effect was not likely due to a reduction of interactions between aggregation-prone intermediates since lysozyme, another small heat stable protein, did not prevent rhodanese aggregation. Rhodanese is a monomeric enzyme of 33 kDa and malate dehydrogenase a homodimer of 35 kDa. Our results suggest that HSP22 could be more efficient in preventing the aggregation of small proteins. It was indeed reported that sHSPs were less effective with large proteins, suggesting that interactions between sHSP and client proteins depend more on the mass ratio rather than the molar ratio [3]. Protection of rhodanese by HSP22 could be of physiological relevance, considering they are major proteins in pea seed mitochondria [56], and the enzyme plays a crucial role in redox homeostasis and sulphur protein biosynthesis [85]. However, even if HSP22rec could partially prevent rhodanese aggregation *in vitro*, enzyme activity was abolished, indicating that other factors are needed to fully protect the enzyme.

HSP22 is a heat-inducible protein, but it is also developmentally expressed during seed maturation, like several other sHSPs [59], which suggests a role of sHSPs in the context of desiccation tolerance. HSP22 was unable to stabilize membranes during desiccation, a function which is likely devoted to the LEA protein (LEAM) that specifically accumulates in pea seed mitochondria [64]. Since our data support the role of HSP22 as a passive chaperone (binding and co-precipitating with unfolding client proteins) during heat stress, it could play a similar role in the context of protein dehydration, which can also cause protein aggregation. However, while the effects of high temperature on protein structure are well established, to our knowledge, nothing is known about the impact of desiccation on the structure of proteins in a highly crowded environment such as the mitochondrial matrix, in which protein concentration can exceed 1 g/mL [86]. HSP22 could also potentially protect proteins of the inner membrane facing the matrix, as was reported earlier for an apple mitochondrial sHSP [41] and more recently for the *Drosophila melanogaster* mitochondrial HSP22 [87]. In the case of desiccation, this would, however, require HSP22 to be located close to the inner membrane, as dehydration would rapidly prevent diffusion of proteins.

Besides acting as a molecular chaperone, HSP22 could also be involved in the amplification of the mitochondrial unfolding protein response (UPRmt) and the mitochondrial proteostasis, as proposed for the *Drosophila melanogaster* mitochondrial HSP22 [24,88]. UPRmt is a complex pathway involved in mitochondrial protein homeostasis [89], which was also demonstrated in plants [90]. Interestingly, one of the six genes encoding a mitochondrial sHSP from cotton was recently proposed to positively control seed germination by interfering with cytochrome *c* and cytbc1 complex biogenesis [91]. The resulting deficiencies of the mitochondrial electron transfer chain would cause an increase of reactive oxygen species (ROS) production, leading to endosperm weakening and subsequent radicle protrusion. Such a mechanism could indeed have a key role in germination control if restricted to the endosperm tissue, otherwise it could compromise energy production, which is essential for seed germination. In the case of HSP22, a similar role cannot be considered, because mature pea seeds do not retain an endosperm layer.

In conclusion, purified HSP22rec was found to form large polydisperse oligomers, which dissociated into monomers at heat shock temperature without detectable structural modifications of the monomer. HSP22 appeared remarkably heat soluble (up to 90 °C). However, upon heating in the presence of other purified client proteins or in complex bacterial or mitochondrial protein extracts, HSP22 was found to co-precipitate with aggregating proteins. Thus, HSP22 appears as a holdase that likely facilitates the subsequent disaggregation and refolding by other molecular chaperones during stress relief or acclimation.

## 4. Materials and Methods

### 4.1. Chemicals

Bis-ANS (1,1’-bis (4-anilino)naphthalene-5,5’-disulfonic acid), cytochrome c, dextran blue, fumarase from porcine heart, lysozyme from egg white, and rhodanese from bovine liver II were purchased from Merck KGaA (Darmstadt, Germany). Malate dehydrogenase from porcine heart was from Roche (Basel, Switzerland) and BSA (fraction V) from Thermo Fisher scientific (Waltham, MA, USA). Purified carboxyfluorescein was a gift from Dirk Incha’s laboratory (Max Plank Institute, Potsdam, Germany). POPC (1-palmitoyl-2-oleoyl-sn-glycero-3-phosphatidylcholine) was purchased from Avanti Polar Lipids (Alabaster, AL, USA).

### 4.2. Mitochondria Isolation and Sub-Fractionation

Purified mitochondria were isolated from 22 h imbibed pea seeds as described in [92]. They were disrupted by 50-fold dilution in a buffer containing 50 mM 3-(N-morpholino)propanesulfonic acid (MOPS) pH 7.5, 1 mM dithiothreitol (DTT), 2 mM ethylenediaminetetraacetic acid (EDTA), and anti-protease cocktail complete mini (Merck KGaA, Darmstadt, Germany). The suspension was subjected to three freeze–thaw cycles (liquid nitrogen/30 °C) to ensure complete lysis. After centrifugation at 100,000 *g* for 1.5 h using an ultracentrifuge and an angular 65Ti rotor (Beckman, Fullerton, CA, USA), the supernatant fraction corresponding to mitochondrial soluble proteins was concentrated by ultrafiltration (Microsep 3K, Pall Laboratory, Westborough, MA, USA). Protein concentrations were determined according to the Bio-Rad Protein assay (Bio-Rad, Hercules, CA, USA).

### 4.3. Expression of Recombinant HSP22

A pUC plasmid carrying a cDNA encoding pea HSP22 precursor (UniProtKB P46254, [52] was used as a template to amplify by PCR a synthetic fragment encoding mature HSP22 with NdeI and BamHI restriction sites for subsequent cloning into the expression vector pET3a. The sequence of the resulting pET-HSP22 was verified by DNA sequencing, and the plasmid finally transformed into BL834pLys *Escherichia coli* strain. The bacteria harboring pET-HSP22 were grown at 37 °C in Luria broth (LB) medium until an optical density (OD 600) of 0.5, and expression was induced by 0.4 mM of isopropyl-beta-D-thiogalactopyranoside (IPTG). After two hours of incubation, bacterial cells were analyzed by SDS-PAGE, revealing a high-level expression of a 19.5 kDa protein that was unambiguously identified as HSP22 after Edman sequencing (LCP, CEA-INSERM, Grenoble).

### 4.4. Purification of Recombinant HSP22

After 2 h of IPTG induction at 37 °C, B834 pLys bacteria overexpressing HSP22 were centrifuged at 4 °C for 10 min at 5000 *g* (JA10 rotor and J2-21M/E centrifuge from Beckman Coulter, Brea, CA, USA). The pellet was resuspended in 20 mM Tris pH 7.4, 1 mM EDTA, and 1 mM phenylmethylsulfonyl fluoride (PMSF), then subjected to one freeze–thaw cycle (liquid nitrogen/37 °C). The suspension was then sonicated using a probe sonicator (Vibracell, Thermo Fisher Scientific, Waltham, MA, USA); five bursts of 1 min (amplitude 80%) with 1 min intervals were applied, keeping tubes on ice. Bacterial extracts were centrifuged for 1 h at 100,000 g at 4 °C using a SW41Ti rotor (Beckman, Fullerton, CA, USA). Supernatants were further subjected to anionic exchange chromatography using a BioLogic LP system (Bio-Rad, Hercules, CA, USA), 5 mL HitrapQ HP columns (GE Healthcare, Chicago, IL, USA), and a flow rate of 3 mL/min. Then, 10 mL of supernatant was diluted in 200 mL buffer A (20 mM Tris pH 7.4, 1 mM EDTA) and filtrated on a 0.22 µm membrane. The column was equilibrated in buffer A before sample loading. The protein elution was performed with 20 mL buffer B (buffer A with 150 mM NaCl) followed by 20 mL buffer C (buffer A with 400 mM NaCl). HSP22 was recovered at both saline steps. Fractions were pooled, concentrated, and desalted with Amicon ultra-15 mL centrifugal filters (50 kDa cut off membrane) from Merck KGaA (Darmstadt, Germany). They were further purified by size exclusion chromatography using an Äkta purifier (GE Healthcare, Chicago, IL, USA) with a Superdex 200 10/300GL (GE Healthcare, Chicago, IL, USA) column and a flow rate of 0.25 mL/min at 20 °C. The elution buffer was 20 mM Tris pH 7.4, 1 mM EDTA, and 150 mM NaCl. The fractions that were enriched in HSP22 and contained few contaminants (as estimated by SDS-PAGE) were pooled, concentrated, and separated by a second size exclusion chromatography in the same conditions as described before. Purified fractions were eventually dialyzed against 20 mM Tris pH 8 and 1 mM EDTA and concentrated as described above. SDS-PAGE analysis of the recombinant purified protein (HSP22rec) revealed a polypeptide with an apparent molecular mass of 19.5 kDa, corresponding to the predicted HSP22 molecular mass (Appendix A). The molecular mass of HSP22 was determined using a calibration curve realized with commercial proteins (size exclusion chromatography calibration kit high molecular weight, GE Healthcare, Chicago, IL, USA) and cytochrome c. Dextran blue was used to determine the void volume. Size exclusion chromatography was also performed at 42 °C with the same column and buffer described above. Samples were heated at 42 °C for 20 min in a water bath and centrifuged before loading in order to remove any insoluble aggregates. The elution buffer was pre-heated at 42 °C in a water bath. A rubber tubing with circulating hot water (heated at 45 °C) was twisted around the column to maintain a constant column temperature of 42 °C during the separation. The elution buffer temperature was controlled at the column entrance and exit. A calibration curve was performed at 42 °C with the same molecular weight markers used for the 20 °C calibration curve (Appendix A).

### 4.5. Electrophoresis and Western Blot Analysis

Proteins were separated by SDS-PAGE according to standard protocols using polyacrylamide gels [10 to 13.5% (*w/v*)], Tris-glycine/SDS buffer [93], and a Miniprotean III apparatus (Bio-Rad, Hercules, CA, USA). For native PAGE, SDS and DTT were omitted from buffers, and samples were not denatured by heating. Following SDS-PAGE electrophoresis, gels were transferred onto a Protran® nitrocellulose membrane (GE Healthcare, Chicago, IL, USA) for 1 h at 100 V in 25 mM TRIS pH 8.3, 192 mM glycine, 0.05% (*m/v*) SDS, and 20% (*v/v*) methanol using a Mini-Transblot system (Bio-Rad, Hercules, CA, USA). Nitrocellulose membranes were blocked for one hour in TBS buffer (10 mM Tris-HCl pH 7.5, 150 mM NaCl) containing 1% (*v/v*) Tween 20. After washing with TBST [TBS containing 0.05% (*v/v*) Tween 20], membranes were incubated with antibodies raised against synthetic peptides (anti HSP22Cterm and antiHSP22Nterm described in [66]) diluted 1:1000 in TBST. Incubations were performed at room temperature for two hours or overnight at 4 °C. After several washes in TBST, membranes were incubated one hour at room temperature with an anti IgG antibody coupled to horse radish peroxidase (Merck KGaA, Darmstadt, Germany) used at a dilution of 1:50,000 in TBST. Immunodetection was performed by incubating the membrane for 5 min in Clarity^TM^ Western ECL Blotting substrate, and chemiluminescence was monitored with a molecular imager Chemidoc^®^ XRS system (Bio-Rad, Hercules, CA, USA).

### 4.6. Circular Dichroism (CD) Spectroscopy

Purified recombinant HSP22 was dialyzed against 5 mM phosphate buffer pH 7.4 and diluted to a concentration of 5 µg/µL in the same buffer; 100 µg of proteins were used for CD measurements. The experiments were carried out in quartz cuvettes with 0.05 cm path length. CD spectra were recorded from 180 to 260 nm with a 0.5 nm step and 2 nm bandwidth using a CD6 circular dichrograph spectropolarimeter (Horiba France SAS, Longjumeau, France) courtesy of UMR BIA (INRA, Nantes, France). Each spectrum (molar ellipticity as a function of the wavelength) was the average of 10 accumulations. Mean residue ellipticities (deg.cm^2^.dmol^−1^.residue^−1^) were calculated stating a molecular mass of 19.5 kDa and 170 residues for HSP22. Data were analyzed with the DICHROWEB software (http://dichroweb.cryst.bbk.ac.uk/html/home.shtml) using the CDSSTR algorithm and reference set data 6 [94,95].

### 4.7. Aggregation Assays

Aggregation assays were performed in 1.5 mL protein low binding microtubes (Eppendorf AG, Hamburg, Germany) with 8 to 25 µg of proteins in 50 µL MOPS 50 mM pH 7.5, DTT 1 mM, and EDTA 1 mM. Samples were heated for 20 min at various temperatures and under continuous agitation (400 RPM) in a Thermomixer incubator (Eppendorf AG, Hamburg, Germany). After 15 min incubation on ice, they were centrifuged for 30 min at 20,000 *g* and 4 °C. Supernatants were mixed with 20% 5 × SDS-PAGE loading buffer. Pellets were dissolved in 50 µL 1 × SDS-PAGE loading buffer. Equivalent proportions of supernatants and pellets were further analyzed by SDS-PAGE.

### 4.8. Enzyme Activity Protection Assay

Bovine liver rhodanese (essentially salt free, purchased from Merck KGaA, Darmstadt, Germany) was diluted in 20 µL of 50 mM MOPS pH 7.5, 1 mM EDTA, and 1 mM DTT to a final concentration of 75 ng/µL alone or with nine-fold molar excess of either recombinant HSP22 or lysozyme (Merck KGaA, Darmstadt, Germany). Then, 10 µL of the mixture were heated for 5 to 20 min at 50 °C and 400 rpm in a Thermomixer incubator (Eppendorf AG, Hamburg, Germany) and then allowed to recover at room temperature for 10 min. The rhodanese activity of heated and native samples was determined by measuring the formation of thiocyanate from cyanide and thiosulfate at 460 nm after 15 min incubation at 25 °C, according to [96].

### 4.9. Bis-ANS Fluorescence Assay

Proteins (0.5 µM final concentration in 120 µL of 100 mM Tris pH 7.5) were incubated for 15 min at different temperatures in a water bath before adding the hydrophobic probe bis-ANS at saturating concentration (37 µM final). Fluorescence emission was immediately recorded for 2 min using a FLUOROstar OMEGA spectrophotometer (BMG LABTECH, Champigny sur Marne, France). Excitation and emission wavelengths were set to 390 and 490 nm, respectively. The fluorescence emitted by the bis-ANS alone was subtracted from all fluorescence measurements.

### 4.10. Membrane Protection Assays

POPC (2.5 mg) was solubilized in chloroform, dried overnight under vacuum, and resuspended in 250 μL of 100 mM carboxyfluorescein (CF) solubilized in TES buffer (10 mM 2-[Tris-(hydroxymethyl)methylamino]-1-ethane sulfonic acid pH 7.4, 0.1 mM EDTA). The suspension was extruded to obtain unilamellar liposomes by using a hand-held extruder (Avestin, Ottawa, ON, Canada) with two layers of 100 nm pore filters. After extrusion, liposomes were further purified on a NAP-5 column (Sephadex G-25; GE Healthcare, Chicago, IL, USA) with TEN buffer (TES buffer supplemented with 50 mM NaCl) to remove unincorporated dye [64]. The lipid concentration was precisely determined; 45 μL of the liposome suspension were solubilized in 0.1% (*v/v*) Triton X100 and then extracted by chloroform in order to remove the CF. Chloroform extracts were dried for 24 h at 110 °C, and lipid dry weights were determined with an ultra-precision balance (AD6, Perkin Elmer, Waltham, MA, USA). Then, 2–5 μg of liposomes were dried overnight in a 96 wells black microplate (Greiner Bio One SAS, Les Ullis, France) alone or in the presence of proteins (final volume of 10 µL). Liposomes were rehydrated in 300 µL of TEN buffer. CF leakage was monitored by fluorescence using a Mithras LB 940 microplate reader (Berthold Technologies Gmbh & Co.KG, Bald Wildbad, Germany). The excitation and the emission light were set respectively at 450 and 535 nm. The CF fluorescence was strongly quenched at the high concentration found inside the intact liposomes but increased when CF was released into the surrounding buffer. After CF fluorescence measurement of rehydrated liposomes, 5 µL of 1% (*v/v*) Triton X-100 were added to induce membrane lysis and release the total CF content. The leakage percentage was calculated according to the following equation: 100 × [(F − F_0_)/(F_T_ − F_0_)], where F_0_ represents the fluorescence of fresh non-dried liposomes, and F and F_T_ represent the fluorescence of rehydrated liposomes before and after the addition of Triton X100, respectively.

### 4.11. Proteome Thermosolubility Assays

After 1 h induction at 30 °C, 100 mL of bacteria overexpressing HSP22 were pelleted by centrifugation (5000 g for 10 min) at 4 °C and suspended in 2 mL of 50 mM MOPS pH 7.4, 1 mM DTT, 1 mM EDTA, and submitted to two freeze–thaw cycles (liquid nitrogen/30 °C). Lysate viscosity was reduced by passing through a 26 G-gauge needle before centrifugation at 4 °C and 14,000 *g* for 15 min. Then, 150 µL of supernatant (2–2.5 µg/µL of proteins) were incubated 20 min at different temperatures in a Thermomixer incubator (Eppendorf AG, Hamburg, Germany) under 400 rpm mixing. Samples were then kept on ice for 15 min and further centrifuged for 30 min at 14,000 *g* and 4 °C in order to separate heat soluble proteins from aggregated proteins. Equal proportions of supernatant and pellet were analyzed by SDS-PAGE. Pea seed mitochondrial soluble proteins were diluted with 50 mM MOPS pH 7.4, 1 mM DTT, and 1 mM EDTA to a concentration of 2 µg/µL and assayed for thermo-solubility as described for bacterial extracts.

### 4.12. Thermotolerance of Bacteria

Ten mL of *E. coli* (B834 pLysS) carrying the pET3a-HSP22 vector were grown at 30 °C until OD at 600 nm reached 0.5. The culture was split into two tubes, one receiving IPTG (0.4 mM final concentration). Induced and non-induced bacteria were further grown for 1 h before stress treatments. Then, 500 µL of bacterial suspension were heated for 30 min at 48 °C in a Thermomixer incubator (Eppendorf AG) with an agitation of 400 rpm. In order to avoid anoxia, bacteria were incubated in 2 mL microtubes sealed with an air-porous membrane lid (LidBac, Eppendorf AG Hamburg, Germany). After treatment, 50 µl of serial dilutions of bacteria were plated in triplicate on LB agar plates. The cell viability was monitored after 16 h incubation at 37 °C by counting the colony-forming units. The survival was calculated using the following formula: Survival (%) = 100 × [(CFU_before stress_ − CFU_after stress)_/CFU_before stress_)].

### 4.13. Statistics

Analysis was performed with the R version 3.3.3 (R Core Team, 2013). Distribution normality and variance homogeneity were tested by Shapiro–Wilk and Barlett tests, respectively. The statistical significance between different means was estimated using t-test or ANOVA test at a level of *p* < 0.05. A Tukey’s test was performed to determine statistically different groups. A Kruskal–Wallis test was used when samples were not normally distributed.

## Figures and Tables

**Figure 1 ijms-21-00097-f001:**
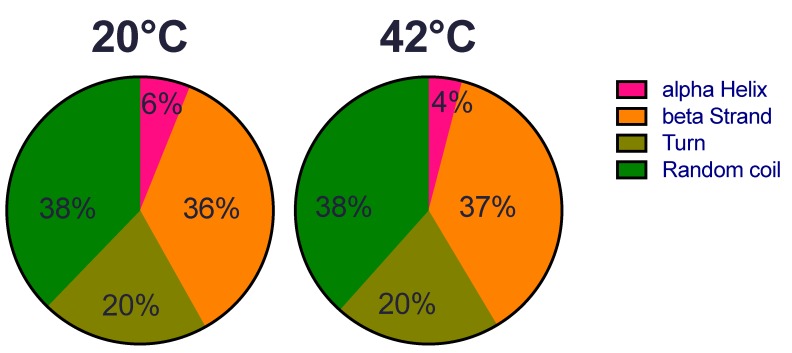
HSP22rec secondary structure determined by circular dichroism at 20 °C and 42 °C. The relative proportion (in %) of alpha helix, beta strand, turn, and random coiled structures was predicted from circular dichroism spectra in the near-UV region using the DICHROWEB software.

**Figure 2 ijms-21-00097-f002:**
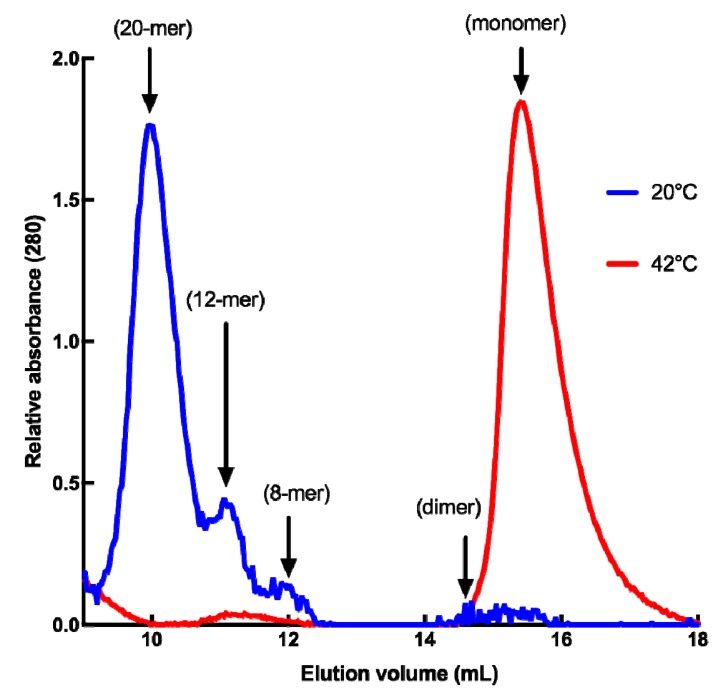
Quaternary structure of the purified HSP22rec at 20 °C and 42 °C determined by SEC (size exclusion chromatography), where 100 µg of recombinant HSP22 were separated by gel filtration, and protein elution was monitored by absorbance at 280 nm. Molecular masses were determined by calibration curves performed with standard proteins at both 20 °C and 42 °C. Corresponding subunit numbers are indicated above peaks.

**Figure 3 ijms-21-00097-f003:**
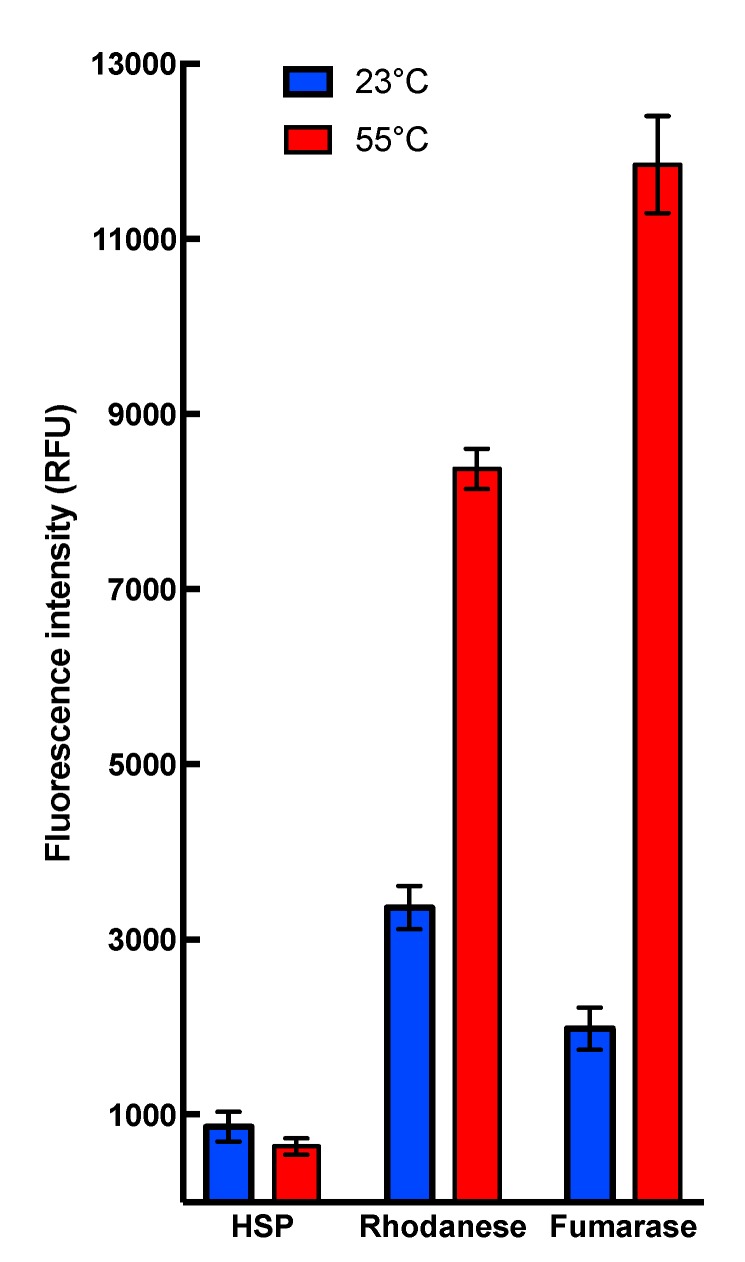
Binding of the hydrophobic probe bis-8-anilinonaphthalene-l-sulphonic acid (bis-ANS) to HSP22rec, rhodanese, and fumarase. Proteins were used at a final concentration of 0.5 µM, calculated on the monomer molecular mass basis for multimeric proteins. They were incubated for 15 min at the indicated temperature (23 or 55 °C). The fluorescent probe bis-ANS was added immediately after heating at a saturating concentration of 37 µM. Each condition was tested at least in triplicate. Fluorescence values are indicated in relative fluorescence unit (RFU). Fluorescence measurements for HSP22 were not significantly different after incubation at 55 °C compared to 23 °C (t-test α= 0.05; *p*-value = 0.071).

**Figure 4 ijms-21-00097-f004:**
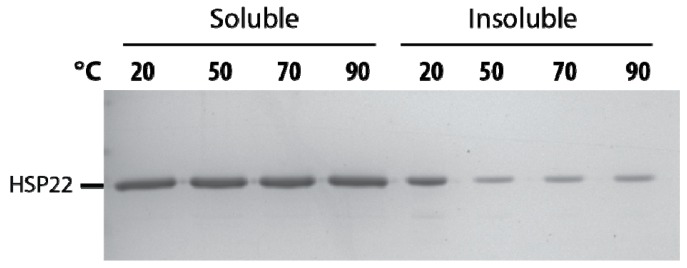
Thermosolubility of the recombinant HSP22. The purified recombinant HSP22 protein (8 µg) was either kept at room temperature (20 °C) or heated for 20 min at the indicated temperatures. After centrifugation, equal fraction volumes of supernatants (soluble fraction) and pellets (insoluble fraction) were analyzed by SDS-PAGE, and proteins were revealed by colloidal blue staining.

**Figure 5 ijms-21-00097-f005:**
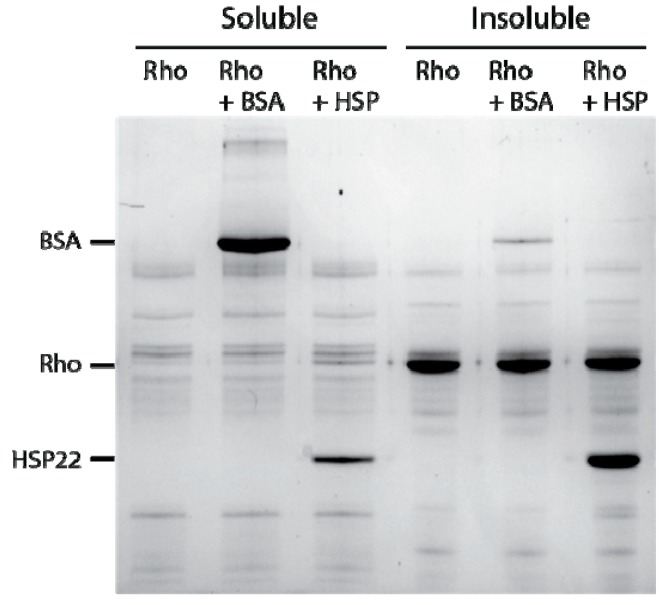
Thermal aggregation of rhodanese at 50 °C in the presence of thermostable proteins. Rhodanese (Rho) (10 µg) was heated 20 min at 50 °C alone or in the presence of bovine serum albumin (BSA) or HSP22rec (HSP) at a 1:1 mass ratio. After centrifugation, equal fraction volumes of supernatants and pellets were analyzed by SDS-PAGE, and proteins were revealed by colloidal blue staining.

**Figure 6 ijms-21-00097-f006:**
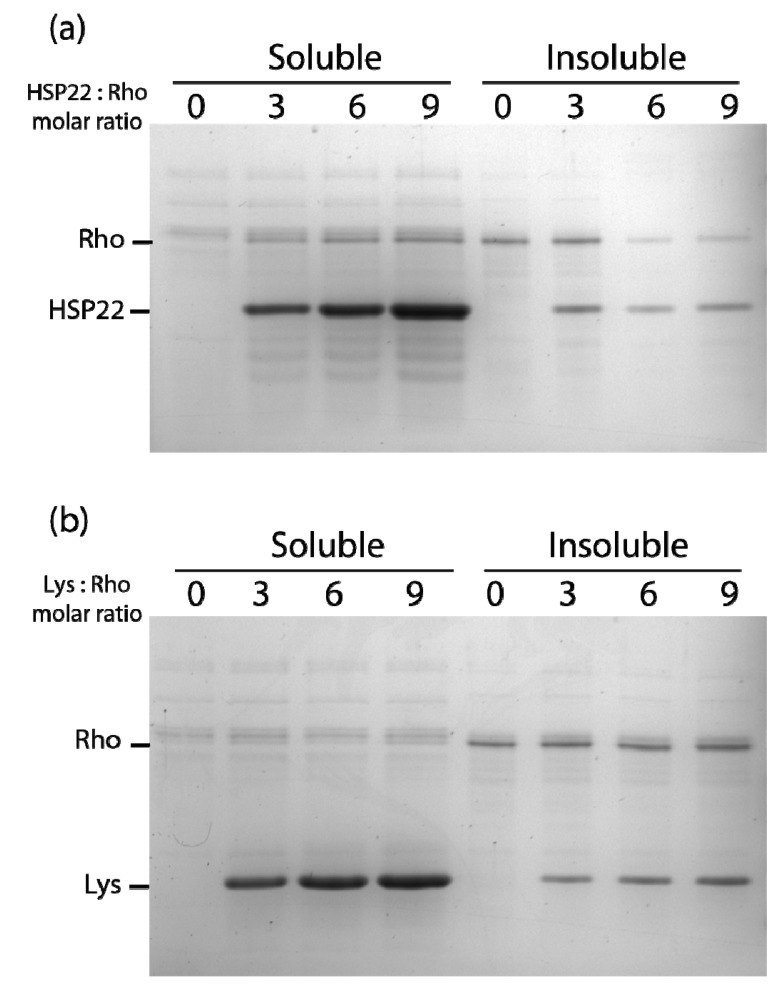
Thermal aggregation of rhodanese in the presence of HSP22rec (**a**) or lysozyme (**b**). HSP22 rec (HSP22) or lysozyme (Lys) were mixed with rhodanese (Rho) using different molar ratio. Samples were heated for 20 min at 50 °C and then centrifuged. Equal fraction volumes of soluble and pellet fractions were analyzed by SDS-PAGE, and proteins were revealed by colloidal blue staining.

**Figure 7 ijms-21-00097-f007:**
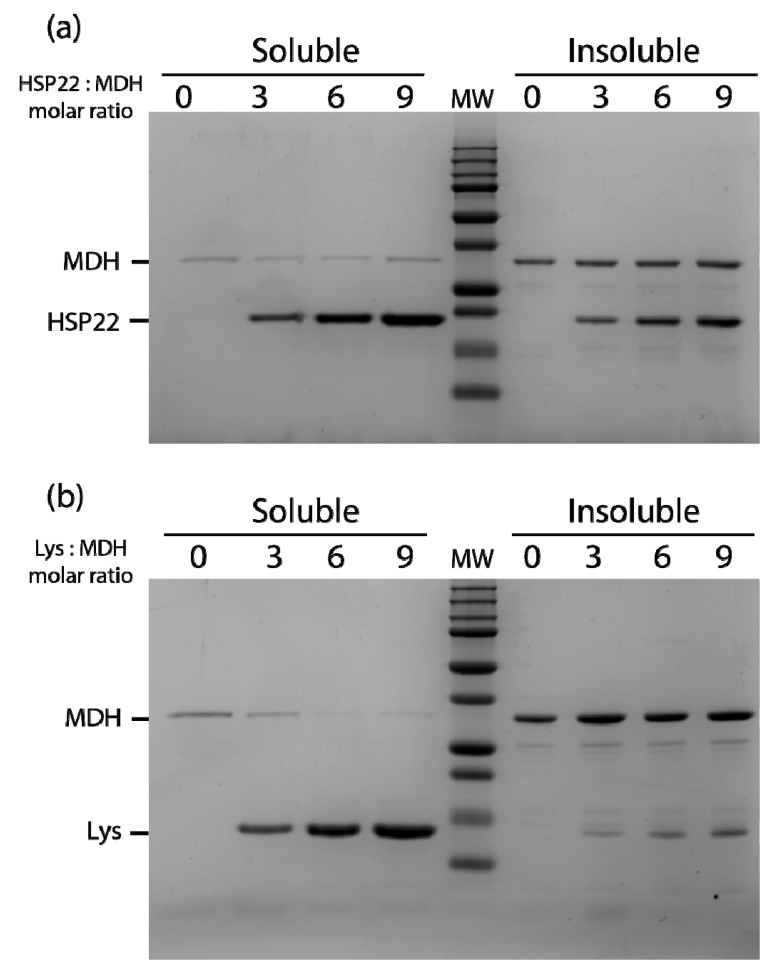
Thermal aggregation of malate dehydrogenase in the presence of HSP22rec (**a**) or lysozyme (**b**). HSP22 rec (HSP22) or lysozyme (Lys) were mixed with malate dehydrogenase (MDH) using different molar ratios. Samples were heated for 20 min at 50 °C and then centrifuged. Equal fraction volumes of soluble and pellet fractions were analyzed by SDS-PAGE, and proteins were revealed by colloidal blue staining.

**Figure 8 ijms-21-00097-f008:**
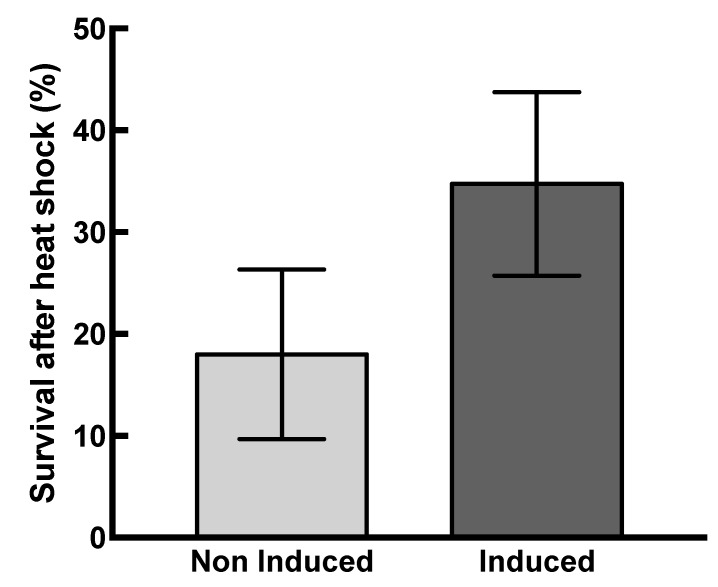
HSP22 overexpression increases bacteria thermotolerance. Induced and non-induced bacterial suspensions were heated for 30 min at 48 °C. Serial dilutions were platted in triplicate on Luria broth (LB) agar plates. The cell viability was monitored after 16 h incubation at 37 °C by counting the colony-forming units (CFU). Survival rate was calculated by the following formula: Survival (%) = 100 × [(CFU_before stress_ − CFU_after stress)_/CFU_before stress_)]. Mean survival percentages and standard deviations (vertical bars) were calculated from ten independent experiments. Black bar: induced bacteria. Grey bar: non-induced bacteria. A significant difference (*p*-value = 0.0004) was found between induced and non-induced bacteria when performing a t-test (α = 0.05).

**Figure 9 ijms-21-00097-f009:**
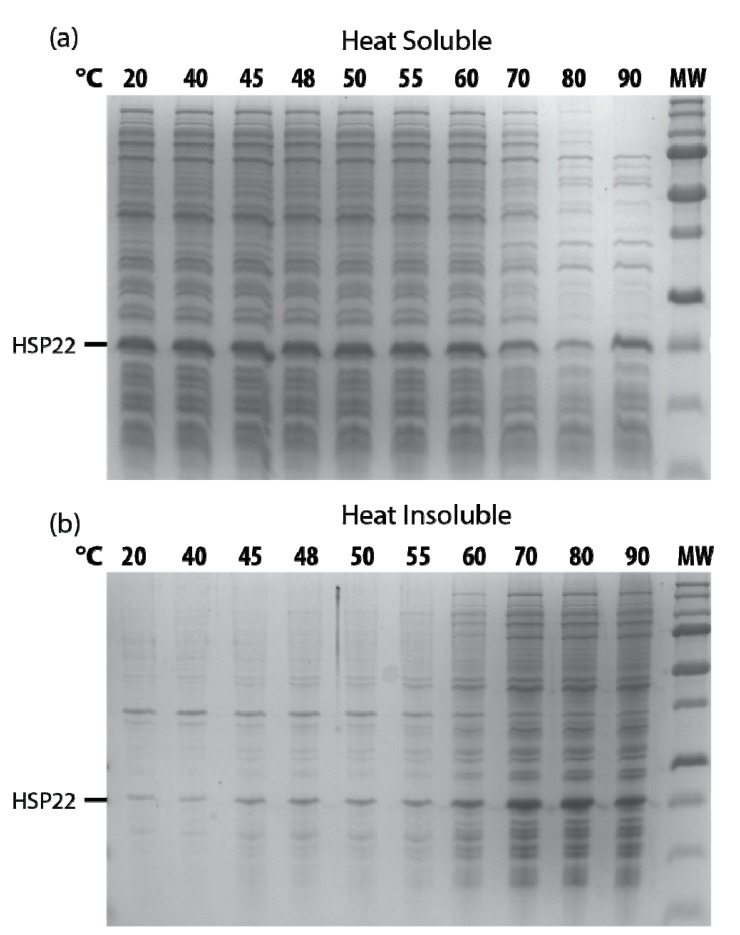
Recombinant HSP22 thermosolubility in *E. coli* extracts as a function of temperature. Soluble protein extracts were obtained from HSP22 overexpressing *E. coli*. Samples (2 µg/µL of proteins) were heated for 20 min at indicated temperatures, kept on ice for 15 min, and then centrifuged. Equal fraction volumes of supernatants (**a**) and pellets (**b**) were separated by SDS-PAGE, and proteins were stained with colloidal blue. MW = 250, 150, 100, 75, 50, 37, 25, 20, 15, KDa (from top to bottom).

**Figure 10 ijms-21-00097-f010:**
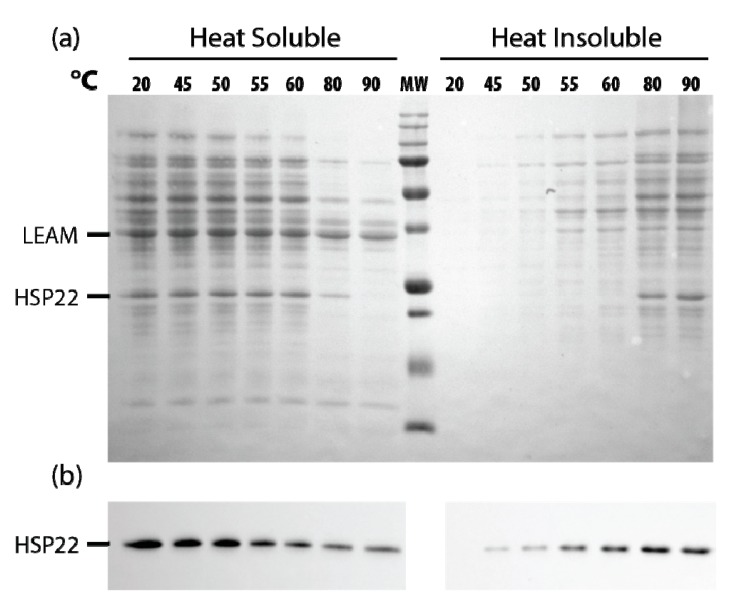
Native HSP22 solubility in pea seed mitochondria extracts as a function of temperature. Seed mitochondria soluble extracts (2 µg/µL of proteins) were heated for 20 min at indicated temperatures and then centrifuged. Equal fraction volumes of supernatant and pellet were analyzed by SDS-PAGE. Gels were then electrotransferred on a nitrocellulose membrane. (**a**) The membrane was stained with Ponceau red, and then (**b**) the membrane was probed with the anti-HSP22-Nterm antibody. MW = 250, 150, 100, 75, 50, 37, 25, 20, 15, KDa (from top to bottom).

**Figure 11 ijms-21-00097-f011:**
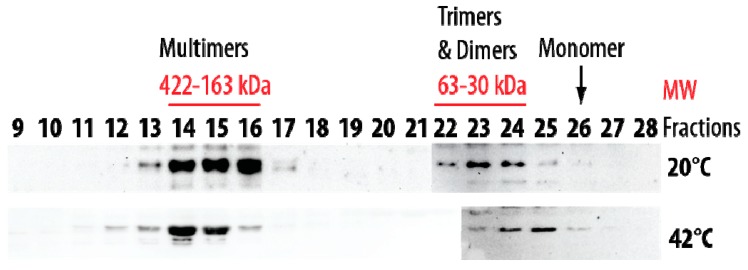
Separation of pea seed mitochondrial soluble proteins by size exclusion chromatography (SEC) performed at 20 °C and 42 °C. Mitochondrial soluble proteins (7 mg) were separated by SEC. For separation at 42 °C, samples were heated for 20 min at 42 °C and then centrifuged prior injection on the column maintained at 42 °C. The collected fractions were analyzed by Western Blot using the anti-HSP22 Cterm antibody. Numbers indicate the molecular mass range of relevant fractions.

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
