# Peer review of "The Mitochondrial Small Heat Shock Protein HSP22 from Pea is a Thermosoluble Chaperone Prone to Co-Precipitate with Unfolding Client Proteins"

_ijms, 2019, doi:10.3390/ijms21010097_

Round 1

Reviewer 1 Report

Macherel and co-workers made an elegant study by investigating the mitochondrial small heat shock protein HSP22 from pea in order to explore the structural features and molecular functions of HSP22 in relation  with stress tolerance. They used the recombinant protein and a wide array of biophysical and biochemical techniques. Overall, this elegant study has clearly shown that HSP22 is a thermosoluble chaperone prone to co-precipitate with unfolded client proteins, i.e. acting as a classical holdase with the ability to facilitate the subsequent disaggregation and refolding by other chaperones  following stress state.

The paper is clearly written, the procedures are described sufficiently, the experiments were carried out correctly and the conclusions drawn are of interest for a diverse audience of scientists. 

Author Response

Thanks for your nice comments.

David Macherel

Reviewer 2 Report

Small Heat Shock Proteins (sHSP) are ubiquitous and abundant proteins participating in protection of living cells against various environmental stresses. To this end they stabilize proteins and lipid membranes including those of mitochondria. One of their important roles in plants is to assure the longevity of seeds, which is important from the agricultural point of view. In the reviewed manuscript the authors analyzed properties of sHSP from pea, belonging to HSP22 family. Using the in vitro assays with purified proteins they showed that HSP22 does not protect against aggregation of heat labile proteins but instead it co-precipitates with them to facilitate their subsequent refolding. They also demonstrated the heat stress-protecting properties of pea HSP22 overexpressed in the heterologous bacterial system. Exposure of bacteria cells to heat progressively precipitated both bacterial proteins and HSP22 even though HSP22 itself is heat-stable. These in vivo results suggested co-precipitation with bacterial proteins as the likely mechanism of protecting bacterial cells against heat stress. They further demonstrated heat-induced precipitation of HSP22 in its native environment – pea seed mitochondria. Altogether, on the basis of their findings the authors postulate co-precipitation of HSP22 with its client proteins as a mechanism of protection against heat stress. Co-precipitation would facilitate refolding of these proteins upon stress withdrawal or their proteolysis.

The manuscript is clearly written and experiments are well designed. The authors discuss obtained results adequately. Methods used are well described. The findings presented in the reviewed manuscript contribute to the elucidation of HSP22 function.

Minor remarks:

page 2 line 21 - “drosophila” should be substituted with italicized “Drosophila melanogaster”, “Drosophila sp.” or with “fruit fly”

page 2 line 48-49 - “Drosophila melanogaster” should be corrected accordingly

page 14 line 18 and 23 - the same issue

page 14 line 19 - “requires” should be substituted with “require”

Author Response

Thanks for your nice comments. We have corrected the fruit fly quotations.

David Macherel